# Determination and Characterization of Novel Papillomavirus and Parvovirus Associated with Mass Mortality of Chinese Tongue Sole (*Cynoglossus semilaevis*) in China

**DOI:** 10.3390/v16050705

**Published:** 2024-04-29

**Authors:** Shuxia Xue, Xinrui Liu, Yuru Liu, Chang Lu, Lei Jia, Yanguang Yu, Houfu Liu, Siyu Yang, Zhu Zeng, Hui Li, Jiatong Qin, Yuxuan Wang, Jinsheng Sun

**Affiliations:** 1Tianjin Key Laboratory of Animal and Plant Resistance, College of Life Science, Tianjin Normal University, Tianjin 300387, China; skyxsx@tjnu.edu.cn (S.X.); liuxinrui0321@163.com (X.L.); 18375410708@163.com (Y.L.); lewchang426@163.com (C.L.); ysy1638@foxmail.com (S.Y.); zeng19985012236@163.com (Z.Z.); lihui258650@foxmail.com (H.L.); 15522897288@163.com (J.Q.); 13040313718@163.com (Y.W.); 2Tianjin Fishery Institute, Tianjin 300221, China; tianjinbohaisuo@163.com (L.J.); yyg555999@163.com (Y.Y.); 18649111793@163.com (H.L.)

**Keywords:** emerging viral disease, *Cynoglossus semilaevis*, papillomavirus, parvovirus

## Abstract

A massive mortality event concerning farmed Chinese tongue soles occurred in Tianjin, China, and the causative agent remains unknown. Here, a novel *Cynoglossus semilaevis* papillomavirus (CsPaV) and parvovirus (CsPV) were simultaneously isolated and identified from diseased fish via electron microscopy, virus isolation, genome sequencing, experimental challenges, and fluorescence in situ hybridization (FISH). Electron microscopy showed large numbers of virus particles present in the tissues of diseased fish. Viruses that were isolated and propagated in flounder gill cells (FG) induced typical cytopathic effects (CPE). The cumulative mortality of fish given intraperitoneal injections reached 100% at 7 dpi. The complete genomes of CsPaV and CsPV comprised 5939 bp and 3663 bp, respectively, and the genomes shared no nucleotide sequence similarities with other viruses. Phylogenetic analysis based on the L1 and NS1 protein sequences revealed that CsPaV and CsPV were novel members of the Papillomaviridae and Parvoviridae families. The FISH results showed positive signals in the spleen tissues of infected fish, and both viruses could co-infect single cells. This study represents the first report where novel papillomavirus and parvovirus are identified in farmed marine cultured fish, and it provides a basis for further studies on the prevention and treatment of emerging viral diseases.

## 1. Introduction

The Chinese tongue sole (*Cynoglossus semilaevis*), of the family Soleidae, in the order Flounder, is naturally distributed in the offshore waters of the Yellow Sea and Bohai Bay. With the breakthrough development of artificial broodstock technology, the breeding scale of Chinese tongue sole has gradually increased, and it has become an important mariculture species in China. The production level reached 2500–3000 tons in 2022, with a current value above 500 million dollars [1].

With the rapid development of intensive aquaculture, disease outbreaks and epidemics have severely constrained the healthy and sustainable development of Chinese tongue sole aquaculture. Bacterial diseases have been described in Chinese tongue sole aquaculture [2,3]. However, research concerning viral diseases associated with Chinese tongue sole is limited. In recent years, emerging viral diseases have threatened the aquaculture of Chinese tongue sole. Li et al. (2014) isolated a betanodavirus (strain CsCN128) from diseased Chinese tongue sole [4], indicating that the fish could be infected by some known viral pathogens. Xiao et al. (2015) have reported that a viral pathogen was responsible for splenic necrotic diseases in Chinese tongue sole [5], but the viral pathogen was not identified. In December 2021, a massive mortality event concerning Chinese tongue sole occurred in Tianjin, China. No parasites were isolated from the diseased fish. Vibrio was found in partially diseased fish, but these bacteria are not associated with the clinical signs observed in the study. A large number of spherical-shaped virus particles were observed in the tissues of the diseased fish using electron microscopy. We attempted to identify it as a known virus pathogen with a similar size and shape, including circovirus [6], hepadnavirus [7], calicivirus [8], piscihepevirus [9], and betanodavirus [10], but no positive result was obtained. These results suggested that a novel viral pathogen was responsible for the emerging disease in Chinese tongue sole.

In the present study, a novel papillomavirus and parvovirus were isolated and identified from diseased Chinese tongue sole. The results of the study will provide a basis for the prevention and control methods of emerging viral diseases.

## 2. Materials and Methods

### 2.1. Fish Samples

Diseased Chinese tongue sole, with a body length of 37–40 cm (17–18 months of age), were collected from a commercial farm in Tianjin, China, in December 2021. The diseased fish were transferred alive in oxygenated bags to the laboratory for diagnosis and detection of the pathogens. Healthy Chinese tongue sole, with a body length of 25–30 cm (11–12 months of age), were obtained from another farm in Tianjin where no history of disease was recorded. All fish were maintained in fiberglass tanks at 21 °C, at 21‰ salinity, with aeration for one week, and no disease symptoms were observed before the experimental pathogen challenge.

### 2.2. Electron Microscopy

Liver, kidney, and spleen tissues of naturally diseased fish were fixed in 2.5% glutaraldehyde and 1% osmium tetroxide for post fixation, then, they were dehydrated, embedded, sliced, and stained with 2% uranyl acetate and lead citrate. All samples were then observed with a transmission electron microscope at 80 KV (JEM1200, Shizuoka, Japan).

### 2.3. Histopathological Observation

Tissue samples of the liver, spleen, and kidney of diseased fish were collected for histopathological observation using hematoxylin and eosin (HE) staining. Tissues were fixed in 4% paraformaldehyde for 24 h, at 4 °C, and washed with Dulbecco’s phosphate-buffered saline (DPBS, Sigma, VA, USA). After being dehydrated using a graded ethanol series to absolute ethanol, an optimum cutting temperature compound (OCT) was used to embed the samples, which were then cut into 8-μm-thick sections using a cryostat (RM2016, Leica, Wetzlar, Germany). The slices were stained with HE and examined using light microscopy (Nikon Eclipse E100, Tokyo, Japan).

### 2.4. Cell Culture and Virus Isolation

FG cells (purchased from the National Cellular Resource Bank and preserved by our laboratory) were cultured in a DMEM medium (GIBCO, New York, NY, USA), supplemented with 10% fetal bovine serum (FBS, GIBCO, USA), at 23 °C, in a 2.5% CO_2_ atmosphere. The tissues from naturally diseased Chinese tongue sole were ground with liquid nitrogen and homogenized with a tenfold volume of DMEM. The suspension was freeze-thawed for three cycles (−80 °C to room temperature), centrifuged at 12,300× *g* for 20 min, at 4 °C, and then filtered through a 0.22-μm membrane. Finally, 1 mL of the filtrate dilution was inoculated onto a confluent monolayer of FG cells in a 25 cm^2^ flask (Corning, New York, NY, USA). The negative (mock infection) control was inoculated with the same volume of DMEM. After adsorption for one hour at 23 °C, 5 mL of DMEM, containing 2% FBS, 100 IU/mL penicillin, and 100 μg/mL streptomycin, was added to the flasks. The cell cultures were incubated at 23 °C and checked daily under an inverted phase contrast microscope (Nikon, Japan) to observe any cytopathic effects (CPE). The supernatant of cells showing CPE was harvested following three freeze-thaw cycles, and centrifugated at 3000× *g*, for 20 min. A subculture of the isolated virus was performed using the methods described above. The virus titer was determined using the 50% tissue culture infective dose (TCID50) method in a 96-well culture plate.

### 2.5. Viral Genome Sequencing and PCR Assays

The tissues of diseased fish were collected, homogenized, and centrifuged to remove cellular debris, and the supernatant was processed through a 0.22-μm filter. The virus was enriched by ultracentrifugation (Beckman Coulter OptimaTM L-80XP, Beckman Coulter, Brea, CA, USA) at 246,347× *g*, for four hours, at 4 °C. The fish nucleic acids were degraded using 200 units of Benzonase (Sigma, USA), six units of TurboTM DNase (Invitrogen, Waltham, MA, USA), and 15 units of RNase I (Invitrogen, USA), and they were incubated at 37 °C for 1 h. RNA and DNA were extracted using a MiniBEST viral RNA/DNA extraction kit (Takara, Shiga, Japan), in accordance with the manufacturer’s instructions. Sequence-independent single primer amplification (SISPA-PCR) was used to analyze the sequences of the unknown viruses. Briefly, viral RNA was converted to cDNA using FR26RV-N (5′-GCCGGAGCTCTGCAGATATCNNNNNN-3′), and then, the viral DNA and the cDNA were processed with Klenow polymerase (Takara, Japan), in the presence of FR26RV-N. Finally, the double-stranded DNA was amplified by PCR, using the primer FR20RV (5′-GCCGGAGCTCTGCAGATATC-3′). The PCR products were used to construct libraries using an NEB Next Ultra DNA Library Prep Kit for Illumina (NEB, Ipswich, MA, USA), in accordance with the manufacturer’s recommendations. Subsequently, the qualified libraries were pooled and sequenced on Illumina platforms using the PE150 strategy, depending on the effective library concentration and the amount of data required. The residual contaminated nucleic acids were filtered by blasting them with the genome of *Cynoglossus semilaevis* [11], using Bowtie 2.2.9 software [12]. The filtered reads were assembled into large scaffolds using metaSPAdes software [13]. Similarity searches were performed using Blast X to find matches with other viruses in the GenBank database. The final genomic sequences were obtained by mapping filtered reads against the scaffolds using Soap de novo 2.0. Multiple primers (Table 1) were designed to amplify the virus genomes, and the genomic sequences were confirmed by Sanger sequencing. The predicted ORFs were analyzed using ORF finder, and a circular genome map of CsPpV was drawn with DNA Plotter. The splicing sites were predicted using the Net-Gene2 Server “http://www.cbs.dtu.dk/services/NetGene2/, (accessed on 8 June 2023)”.

### 2.6. RNA Extraction and L1 Gene Sequencing

The complete L1 gene was obtained from the genomic DNA of CsPaV and inserted into pEGFP-C1 to construct the plasmid, pEGFP-C1-L1. Transfections were performed using Lipofectamine 3000 (Invitrogen), in accordance with the manufacturer’s instructions. The total RNA was extracted from viral infected tissues, FG cells, and transfected FG cells using a TRIzol Reagent. First-strand cDNA synthesis was performed using M-MLV Reverse Transcriptase (Promega, Madison, WI, USA) and specific primers for the L1 gene. The cDNA samples were subjected to PCR, and the PCR products were sequenced and blasted with the L1 gene.

### 2.7. Phylogenetic Analysis

The L1 amino acid sequences of papillomavirus and NS1 amino acid sequences of parvovirus are usually used to analyze phylogeny [14,15]. A similarity comparison was conducted by comparing BLASTn and BLASTp against the NCBI database. The deduced amino acid sequences were analyzed using Expasy “https://web.expasy.org/translate/, access on 21 July 2023”. Multiple sequence alignment was performed using the Muscle package with default parameters. The L1 amino acid sequences of the identified fish papillomavirus, and NS1 amino acid sequences from other representatives of the subfamilies Parvovirinae and Hamaparvovirinae, were employed using MEGA 10.0 and the maximum likelihood method, with a Poisson substitution model and a gamma distribution of rates. The node support values were obtained with 100 bootstrap iterations.

### 2.8. Fluorescence In Situ Hybridization (FISH)

To investigate the location of the mRNAs of CsPaV and CsPV in the infected fish, the Paraffin-SweAMI-double FISH method was employed, with FAM-conjugated probes for the CsPaV L1 and Cy3-conjugated probes, for CsPV VP gene sequencing (Table 1). Spleen samples from diseased fish were fixed in 4% paraformaldehyde (PFA, Sigma, USA), embedded in paraffin, and sectioned (4 μm) at −20 °C (RM2016, Leica, Germany). The sections were heated at 62 °C for 2 h, followed by dewaxing and dehydration with ethanol, then, they were digested with proteinase K (20 μg/mL). The sections were hybridized overnight at 40 °C with conjugated probes. After washing with the hybridization solution, the corresponding branch probes were added for hybridization. Then, the signal probes at a dilution ratio of 1:400 were added and hybridized overnight at 40 °C. Next, a DAPI solution was added to the sections to counterstain the nuclei. DAPI fluoresces blue via UV excitation at a wavelength of 330–380 nm, and an emission wavelength of 420 nm; FAM fluoresces green (L1 probe of CsPaV) at an excitation wavelength of 465–495 nm, and an emission wavelength of 515–555 nm; and CY3 fluoresces red (the VP probe of CsPV) at an excitation wavelength of 510–560 nm, and an emission wavelength of 590 nm. All sections were examined using an inverted fluorescence microscope (Nikon Eclipse TI-SR, Tokyo, Japan).

### 2.9. PCR Detection and TaqMan Probe-Based Quantitative PCR

DNA was extracted from a purified virus, infected FG cells, and tissue samples using a DNA viral kit (Omega, Berlin, Germany), in accordance with the manufacturer’s instructions. The E1 gene sequence of CsPaV, and the VP gene sequence of CsPV, were used to design primers (PaVF: 5′-ATGAGTGAGTTCCTATCTAT-3′, PaVR: 5′-TGAATGGCGTGCTCTAAT-3′; PVF: 5′-ATGGGACTTACACGAAT-3′, PVR: 5′-GTTATTTTGTTGCCTTG-3′) for virus detection. Subsequently, the virus DNA samples were subjected to PCR (95 °C for 5 min, followed by 35 cycles of 94 °C for 1 min, 58 °C for 1 min, and 72 °C for 1 min, then a final extension at 72 °C for 10 min) that produced 654 bp and 1498 bp PCR products for CsPaV and CsPV, respectively. Inverse PCR was performed to evaluate the circular genome of CsPaV. Primers that used inverse PCR were PaVIPCRF: 5′-GGCGATATAGTAAGGATGACCAAG-3′ and PaVIPCR: 5′-TGAATGGCGTGCTCTAATACC-3′. The virus DNA samples were subjected to PCR (95 °C for 3 min, followed by 35 cycles of 94 °C for 10 s, and 48.2 °C for 10 s, and 72 °C for 1 min, then an extension at 72 °C for 7 min), which produced a 2046 bp PCR product.

A TaqMan probe-based quantitative PCR was developed and used for the detection and quantification of CsPaV and CsPV in infected Chinese tongue sole tissue samples. Virus genomic DNA was extracted using a DNA viral kit, as mentioned above. The primers (L1F: 5′-AATCAGATCAAGAAGAAGACCCA-3′, L1R: 5′-CCTCATTATCATAGATGGTGCC-3′; VPF: 5′-CCCAATGGAAACCCAAGACTC-3′, VPR: 5′-CCGCTAGTATCAGGTTTTCCG-3′) and probes (L1Probe: FAM-CTGGAACACCGAACAGTGACTATGGC-TAMRA; VPProbe: FAM-CCAAAAGAAAAAGAGACCCCGACGAC-TAMRA) were designed using Primer 5.0. A Premix Ex TaqTM (Probe qPCR) (TaKaRa, Japan) was used to perform qPCR, in accordance with the manufacturer’s instructions. Each reaction consisted of 1 μL DNA template. The assay was performed at 95 °C for 2 min, followed by 40 cycles of 95 °C for 5 s, and 60 °C for 30 s. Each test was performed in triplicate, together with a no template control in each run. To determine the copy numbers of the viruses, the standard curves of the TaqMan qPCR assays were generated using 10-fold dilutions of the plasmid containing the DNA fragments of the E1 gene from CsPaV and the VP gene from CsPV. The virus copies were calculated using the Ct values and the formula (Log virus copies/μL = −0.294 Ct + 12.216 for CsPV and Log virus copies/μL = −0.296 Ct + 12.486 for CsPaV) generated above.

### 2.10. Animal Infection Experiments

The challenge experiments were performed using 96 healthy Chinese tongue sole. Prior to the experiment, 20 fish were randomly selected and detected for CsPaV and CsPV by qPCR, and the results were determined to be negative. Forty-eight fish in the test group were challenged using an intraperitoneal (IP) injection, with 0.3 mL of the homogenate of diseased fish tissues, prepared as described above, and filtered through a 0.45-μm filter. An additional 48 fish in the control group were injected using an IP, with 0.3 mL of Dulbecco’s PBS. All fish were kept in tanks, supplied with aerated water at 21 °C, at a salinity of 20‰ during the infection experiment. Clinical signs and mortality were checked daily.

## 3. Results

### 3.1. Diseases and Pathological Features

A mass mortality event concerning farmed Chinese tongue sole occurred in December 2021, in Tianjin, China. The water temperature and salinity ranged from 17 °C to 20 °C, and from 16‰ to 20‰, at the focal sites. The disease outbreak lasted for about one month (from 16 December 2021 to 16 January 2022), and the cumulative mortality rate reached 80.7% (Figure 1B). Clinical symptoms observed included punctuate hemorrhages in the liver, a swollen spleen, and kidneys filled with white nodules (Figure 1(A-1,A-2)).

In diseased fish, the liver, spleen, and kidney appeared most severely affected. The liver showed hepatocytes, oedema, and necrosis (Figure 1(C-1)). The spleen appeared to be severely affected, showing vacuolation and necrosis (Figure 1(C-2)). In the kidney, renal tubular atrophy and necrosis occurred, accompanied by a reduction in lymphocytes, which were sparsely arranged (Figure 1(C-3)).

### 3.2. Electron Microscopy

Viral particles were observed in the liver (Figure 2A), kidney (Figure 2B), and spleen (Figure 2C,D). Electron microscopy revealed numerous spherical non-enveloped viruses scattered in the cytoplasm and nuclei of cells. Moreover, large, nearly circular inclusions, filled with viral particles, were observed in the cytoplasm near the nucleus.

### 3.3. Virus Isolation

Changes in cell morphology and CPE were examined three days post infection (dpi) (Figure 2E). Typical CPE, including cell shrinkage, rounding, and cytoplasmic vacuolization, were observed at 5 dpi and the second cell culture passage (Figure 2F). The FG cells were rounded and floating in the medium at 8 dpi at the third cell culture passage (Figure 2G). No CPE was observed in uninfected FG cells (Figure 2H). The TCID50 was determined as 10^−10.75^/mL, at the second cell culture passage, at 8 dpi.

### 3.4. Genome Sequencing

SISPA-PCR, combined with high-throughput sequencing, was used to analyze the full viral genome sequence. No results were obtained from the RNA template. Unexpectedly, two virus-associated proteins belonging to the Papillomaviridae and Parvovidae families were found after comparing the filtered sequences with other viral proteins in the GenBank database using BLASTX. Two complete genomes were assembled using Soap de novo 2.0, and the sequences of the genomes were confirmed by overlap PCR or inverse PCR followed by Sanger sequencing.

The complete genome of CsPaV comprised 5939 bp and had a GC content of 36.95%. The genome contained four predicted protein-coding regions that were typical backbone proteins of papillomavirus (E1, E2, L1, and L2), lacking any of the oncogenes (E5, E6, and E7) [16] (Figure 3A).The results of the inverse PCR revealed that CsPaV had a circular genome and that it was not an endogenous viral element in the genome of the host, *Cynoglossus semilaevis* (Figure 3B). The E1 protein had a DNA helicase (aa 314–561) at the C terminal [17], and five cyclin A interaction motifs (RXL; aa 90–92, aa 186–188, aa 383–385, aa 469–471, aa 477–479). The E2 protein contained a transactivation domain (aa 84–171) and a viral DNA binding domain (aa 213–288) which were identified from other papillomaviruses [18]. Moreover, two small open reading frames (sORF1 and sORF2), that showed no similarity to other proteins in the database, were found at the region between the end of the L1 ORF and the beginning of the E1 ORF. The sORF1 encoded 70 amino acids and had a pRb pocket (LXCXE motif). The sORF2 encoded 71 amino acids and had a heavy metal binding motif (CXCX4CXC). The PDZ-binding motif that is correlated with oncogenic potential [19] was absent in both sORF1 and sORF2. Additionally, an E1 binding site (E1BS; ATNGTTN3AAGNAT) was found at the non-coding area before sORF1 (nt 5498–5512). Interestingly, the encoding region of CsPaV L1 was composed of two ORFs (L1a and L1b), with L1a overlapping by 195 nt with L1b, as predicted by ORF finder. The unusual structure suggested that the L1 protein might be expressed using a spliced mRNA. To confirm this, the splicing sites were predicted using the Net-Gene2 Server. The results showed that donor and acceptor splice sites were found at nt 617 (confidence 0.83) and nt 683 (confidence 0.97), respectively. To verify the predictions, the total RNA of the CsPaV-infected tissues, FG cells, and transfected FG cells was extracted, and then RT-PCR was conducted to amplify the L1 gene. The PCR products were sequenced and BLASTed with the sequence obtained from the DNA template. The results showed that a 67 bp segment, with a high AT content (81.6%), was spliced in the cDNA template, and the splicing sites were at nt 617 and nt 683, as expected (Figure 3C). The complete genome sequence of CsPaV was deposited in GenBank (accession no. OQ865369).

A novel parvovirus (CsPV) was also found in the diseased fish. The nearly complete genome was 3663 bp, and it contained three ORFs encoding the well-known NS1 and VP proteins of parvoviruses, and an unknown ORF3. The VP gene overlapped by 32 nt with the NS1 gene; the unknown ORF3 overlapped by 103 nt with the VP gene (Figure 3D). The NS1 was predicted to be 891 bp (296 aa) in length, which was significantly shorter than other parvoviruses in the subfamily Parvoviridae (569–651 aa). NS1 contained a helicase domain (aa 120–218), including the conserved ATP- or GTP-binding Walker loop motifs, a Walker A loop aa motif (GxxxxGKT/S; 127GPPSTGKT137), and the Walker B (xxxxEE; 168LIWMEE175), Walker B’ (KxxxxGxxxxxxxK; 185KAVAGGTDVLIDVK200), and Walker C (PIxIXXN; 210PVIWTTN218) aa motifs [14]. The HuH motif that is widely present in the NS1, was absent in CsPV [20]. The VP protein was 631 aa in length, similar to other members of the subfamily Parvovirinae (537–781 aa). Similar to other identified fish parvoviruses [21,22], the phospholipase A2 motif, that is widely present in the VP protein of many parvoviruses [23], was not found in CsPV. In addition, an unknown ORF3 encoding 182 aa was found, and it showed no significant matches with other parvoviruses after using BLASTP. The genome sequence of CsPV was deposited in GenBank (accession no. OR795057). The sequencing reads were deposited into the Sequence Read Archive (SRA) under accession number PRJNA1040012.

### 3.5. Phylogenetic Analysis

Phylogenetic analysis, based on the amino acid sequence of the L1 protein (Figure 4A), revealed that the L1 protein of CsPaV shared the highest similarity (53.45%) with Papillomaviridae sp. Haddock_c6033. According to the genus demarcation criteria of the International Committee on Taxonomy of Virus (ICTV) for family Papillomaviridae (viruses with >70% identity in the L1 sequence belong to the same species; those with >60% identity belong to the same genus), we proposed that CsPaV belonged to a new genus of this group. For CsPV, a phylogenetic tree was constructed based on the NS1 protein amino acid sequences of 20 representative parvoviruses in the subfamilies Parvovirinae and Hamaparvovirinae, which are included in the family Parvoviridae (Figure 4B). CsPV clustered with Parvovirinae sp. isolated fi102par1 [24] and zander/M5/2015/HUN [22], and shared 51.15% and 34.85% similarity, respectively. Based on the ICTV Parvoviridae Study Group, all parvoviruses in a genus should be monophyletic and encode NS1 proteins that are >30% identical at the amino acid sequence level [25]; we speculated that CsPV belonged to a new genus as Parvovirinae sp. isolated fi102par1 and zander/M5/2015/HUN in the subfamily Parvovirinae.

### 3.6. FISH Detection

The mRNA expression of CsPaV and CsPV in the infected fish was analyzed using the Paraffin-SweAMI-double FISH method, and it is shown in Figure 5A–C. The transcripts of the two viruses were located mostly in the cytoplasm (red/green), and fewer were found in the nucleus (purple/cyan). Moreover, CsPaV and CsPV could be detected in the cytoplasm (yellow) and nucleus (white) of single cells.

### 3.7. Tissue Distribution of CsPaV and CsPV

The livers, spleens, kidneys, intestines, gills, and muscles of naturally infected Chinese tongue soles were collected, and the virus load (viral genomic copies) was analyzed via probe-based quantitative PCR. The results showed that both viruses had high loads in the examined tissues. CsPaV had the highest loads in the kidney (10^7.02 ± 0.02^/mg), followed by the intestine (10^6.96 ± 0.03^/mg), and almost equal loads were observed in the liver, spleen (10^6.57 ± 0.01^/mg), (10^6.55 ± 0.01^/mg) and muscle (10^6.57 ± 0.05^/mg), with the gills (10^6.23 ± 0.03^/mg) containing the least amount of virus. For CsPV, the loads in the tissues were significantly lower than those of CsPaV. The virus genomic copies of CsPV were much higher in the spleen (10^5.17 ± 0.04^/mg), kidney (10^4.81 ± 0.01^/mg) and liver (10^4.68 ± 0.06^/mg) than in the gills (10^4.45 ± 0.08^/mg), intestine (10^4.37 ± 0.02^/mg), and muscle (10^4.37 ± 0.01^/mg) (Figure 6A).

### 3.8. Animal Experiments

Clinical signs of the fish in the experimental group included feeding disorders and decreased homeostasis, slightly swollen spleens and kidneys, and the livers were losing blood. We observed white nodules in the spleen and kidneys of only some of the infected fish during the last two days of the infection experiment, which did not fully correspond with the signs of the naturally diseased fish. The cumulative mortality reached 100% at 7 dpi (Figure 6B). All of the fish in the control group remained asymptomatic. Two fish were randomly selected from the experimental and control groups at 5 dpi to detect the CsPaV and CsPV via PCR. The results showed that samples in the experimental group were PCR positive for CsPaV and CsPV, whereas samples from the control group were negative (Figure 6C).

## 4. Discussion

Chinese tongue sole is an important mariculture species in China. In recent years, an emerging viral disease has threatened the culture of this fish. Typical clinical signs in naturally diseased fish are swollen spleens and kidneys, which are filled with white nodules. However, in this artificial infection experiment, we observed slightly swollen spleens and kidneys, and white nodules were only observed in the spleens and kidneys of some fish. We analyzed the possible reasons for this result, and found that the IP injection inoculates a large number of virus particles into the fish in a short period of time, which can lead to the acute death of the fish, whereas in natural culture conditions, the virus enters the fish through the skin, gills, or intestines by contacting the virus-infected fish. The appearance of typical signs of disease requires a process. We will change the infection method (such as immersing infection) to observe the disease signs in the fish of the experimental group in future works. Nevertheless, we believe that the results of the infection experiments prove that both viruses can cause massive mortality in Chinese tongue sole and are pathogenic agents.

A novel parvovirus and papillomavirus were isolated and identified from diseased Chinese tongue sole. Parvoviruses are small, non-enveloped viruses with linear single-stranded DNA [25]. To date, a total of five fish parvoviruses have been identified. Parvoviruses from fish in the Lhasa River [24], Zander PV [22], and ichthyic PV [21] were obtained from fish with no signs of disease. Therefore, it remains unclear whether the three viruses are disease-causing agents. TiPV was demonstrated to be the causative agent in tilapia, and it was found to belong to the genus Chapparvovirus, subfamily Hamaparvovirinae [20]. CsPV belonged to a new genus in the subfamily Parvovirinae, of the family Parvoviridae. We suggest that CsPV is the first disease-causing parvovirus identified from marine cultured flatfish. The virus may reflect an evolutionary adaptation to high salinity, thereby broadening the present host range of parvoviruses.

The second novel disease-causing virus was CsPaV. Papillomaviruses are small, icosahedral, non-enveloped viruses with circular double-stranded DNA [15]. Phylogenetic analysis revealed that CsPaV clustered with other identified fish papillomaviruses, suggesting a possible monophyletic origin of the group. To date, papillomaviruses have been found in eight species of fish, including gilthead sea bream (*Sparus autata*) [26], wels catfish (*Silurus glanis*) [27], black sea bass (*Centropristis striata*), emerald notothen (*Trematomus bernacchii*), haddock (*Melanogrammus aeglefinus*) [28], rainbow trout (*Oncorhynchus mykiss*), red snapper (*Lutjanus campechanus*) [29], and fish from the Lhasa River [24]. Among all the determined fish papillomaviruses, only SaPV1 and SgPV1 have been reported to be agents causing typical signs of diseases in fish such as papilloma-like lesions or papilloma-like epidermal hyperplasia of the skin. Such signs are consistent with the typical signs of papillomavirus infections, due to epitheliotropic viruses inducing infections in the stratified squamous epithelia of the skin and mucosal membranes [30]. However, we observed tumor-like signs in the kidney and spleen tissues of diseased Chinese tongue sole. Previous studies have shown that papillomaviruses could bind to heparin sulfate proteoglycans on either the epithelial cell surface or the basement membrane through interactions with the L1 protein [31,32], and that its replication cycle is tightly coupled to the differentiation state of infected squamous epithelia [33]. We speculated that CsPaV may have evolved new strategies to bind with and enter the cells, then replicate in the cells of visceral tissues, due to the altered histophilicity.

Unusual genome organization was found in CsPV and CsPaV. There are four typical genome structures, containing two major ORFs, encoding NS1 and VP, in the family Parvoviridae “https://ictv.global/report/chapter/Parvoviridae/Parvoviridae, (accessed on 22 December 2023)”. The genome of CsPV, in addition to the ORFs of NS1 and VP, contained an unknown ORF3, encoding 182 amino acids at the 3′ end, with no homology to any nucleotide or amino acid sequences in GenBank. Moreover, the NS1 gene and the VP gene, as well as the VP gene and the unknown ORF3, overlapped; this pattern is uncommon in the subfamily Parvovirinae. In a recent report, an unknown 125-aa-long protein at the 5′ end was found in a parvovirus (zander/M5/2015/HUN) identified from fecal specimens of zander, and the NS1 overlapped with VP by 361 nt, while VP and the unknown protein were separated by 42 nt [22]. Moreover, a parvovirus identified from fish in the Lhasa River had a similar genome organization with CsPV, and it contained an unknown 145-aa-long protein at the 3′ end [24]. To date, only these three fish parvoviruses, belonging to unassigned genera, subfamily Parvovirinae, have demonstrated uncommon genome structures. Moreover, CsPV is the first parvovirus identified from diseased mariculture fishes, suggesting an evolutionary adaptation to high salinity, and it broadens the present host range of parvoviruses. We suggest that it is necessary to further research the genetic diversity of fish parvoviruses and explore their evolutionary relationships. As with other fish papillomaviruses, CsPaV contains the minimal backbone genes (E1, E2, L2, and L1), and is lacking any of the oncogenes (E5, E6, and E7) [16]. Of note, the L1 protein was expressed with a spliced mRNA, and a 67 bp segment with a high AT content acted as an intron. Virus introns are rarely found in small viruses, although they appear in some viruses with large genomes [34,35,36]. Lopez-Bueno et al. (2016) [26] reported that the SaPV1 L1 protein was expressed with a spliced transcript. Although these are the only two instances of splicing events occurring in the L1 encoding ORF of a fish papillomavirus, we speculate that it may be a common strategy employed to infect fish with SaPV and CsPaV, since they are both disease-causing agents and have been shown to co-infect with other viruses. Further studies are needed to address this hypothesis.

In the present study, CsPaV and CsPV were isolated and analyzed simultaneously from the diseased Chinese tongue sole, and the FISH results revealed that the transcripts of the two viruses could be detected in single cells of the infected tissues, hinting that the emerging viral disease was caused by a co-infection of the two viruses. Viral co-infections are common in nature [37,38]. However, studies and available data on viral co-infections in fish aquaculture are limited. In recent years, the co-infection of papillomavirus with other viruses has occurred in fish aquaculture, such as SaPV1 co-infected with iridovirus and polyomavirus [26], and SgPV1 co-infected with herpesvirus [27]. However, as in these reports, we were unable to clarify which of the viruses was the dominant one that contributed to the disease; this is because it was difficult to separate and purify the viruses from the infected tissues or FG cells. We believe that constructing infectious clones of the two viruses could shed light on this issue. It was reported that an adeno-associated virus (AAV), a type of parvovirus, can interact with human papillomavirus (HPV) and reduce the risk of developing HPV-16-induced cervical cancer [39,40]. HPV16 was shown to play a significant role in enhancing AAV replication, and AAV replication could interfere with the replication and oncogenic potential of papillomavirus [41,42]. Whether CsPaV and CsPV have similar interaction patterns is unknown, and thus, the mechanism requires further research.

## 5. Conclusions

In conclusion, the novel papillomavirus (CsPaV) and parvovirus (CsPV) were isolated and characterized from the diseased Chinese tongue sole. The two viruses appear to be emerging viral pathogens of farmed Chinese tongue sole. Further research is needed to elucidate the mechanisms of virus co-infection and to determine methods for the prevention and control of the emerging disease.

## Figures and Tables

**Figure 1 viruses-16-00705-f001:**
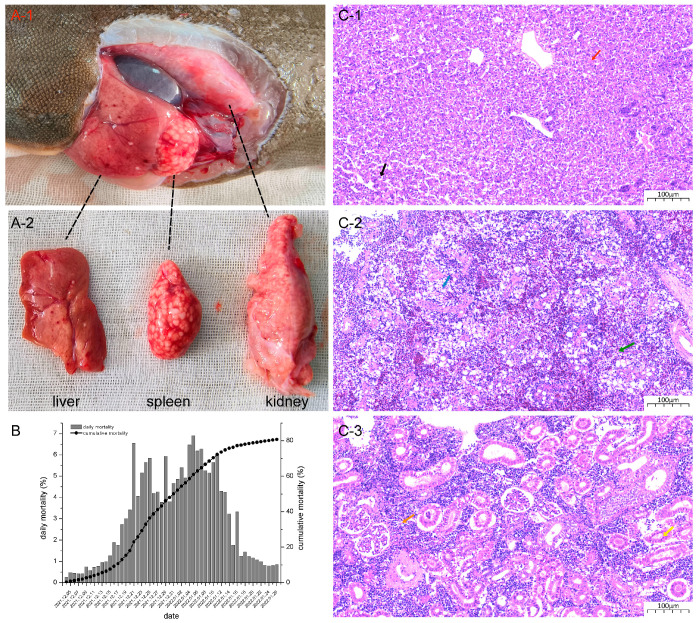
Clinical signs of diseased Chinese tongue soles. (**A-1**,**A-2**) Anatomical symptoms including a swollen kidney and spleen filled with white nodules, and punctate hemorrhages on the liver; (**B**) statistics on daily and cumulative mortality; (**C-1**) diseased fish liver showing oedematous (black arrow) and necrotic (red arrow) hepatocytes; (**C-2**) diseased fish spleen showing different levels of vacuolation (blue arrow) and necrosis (green arrow); (**C-3**) renal tubular atrophy and necrosis (yellow arrow), and sparsely arranged lymphocytes (orange arrow) in a diseased fish kidney.

**Figure 2 viruses-16-00705-f002:**
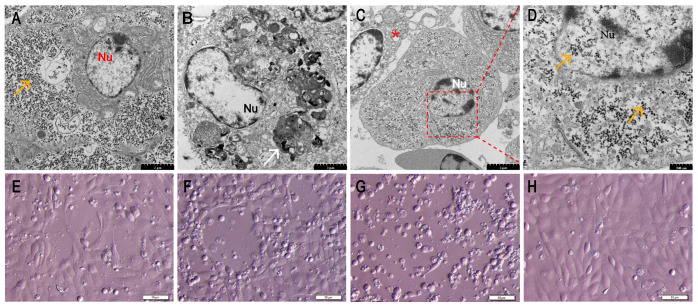
Identification of the causative pathogen in diseased Chinese tongue soles using transmission electron microscopy and virus isolation. (**A**) Liver: a large amount of virus particles were presented in the cytoplasm (orange arrow). Nu: nucleus (Bar, 2 μm). (**B**) Kidney: large, nearly circular inclusions (white arrow) in the cytoplasm near the nucleus (Bar, 2 μm). (**C**) Spleen: virus particles presented in the cytoplasm and nucleus; necrotic and lysed cells were observed (asterisk) (Bar, 2 μm). (**D**) Higher magnification of the area bounded by the red rectangle in panel C, showing magnified virus particles (orange arrow) (Bar, 500 nm). (**E**) FG cells infected with CsPaV and CsPV at passage 1, for 3 days. (**F**) FG cells infected with CsPaV and CsPV at passage 2, for 5 days. (**G**) FG cells infected with CsPaV and CsPV at passage 3, for 7 days. Typical CPE was observed (Bar, 50 μm). (**H**) Normal FG cells.

**Figure 3 viruses-16-00705-f003:**
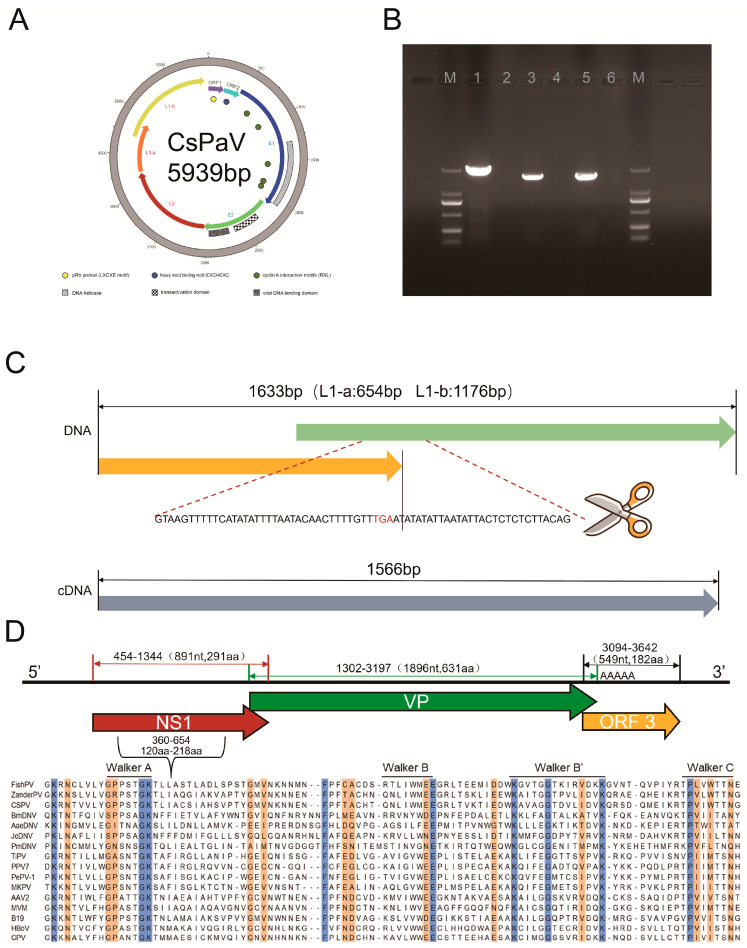
Genomic analysis of CsPaV and CsPV. (**A**) Circular graphs of the genome of CsPaV. Outer scales are numbered in kilobase pairs in a clockwise direction. The predicted E1, E2, L2, and L1 genes are shown in different colors with arrows. (**B**) Inverse PCR on the extracted genomic DNA of CsPaV infected (lane 1) and uninfected (lane 2) spleen tissues of fish; RT-PCR on RNA, extracted from CsPaV infected (lane 3) and uninfected (lane 4) FG cells; PCR on genomic DNA, extracted from CsPaV infected (lane 5) and uninfected (lane 6) FG cells. (**C**) Diagram showing CsPaV-L1 containing two ORFs (L1a and L1b), with the spliced sequence (67 bp) indicated. (**D**) Genome organization of CsPV (The NS1, VP, and ORF3 are shown in different colors) and alignment of conservative helicase domains, including the Walker A, B, B’, and C box of the NS1 protein.

**Figure 4 viruses-16-00705-f004:**
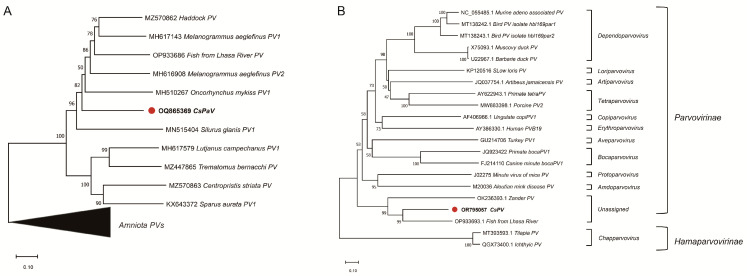
Phylogenetic analysis of CsPaV and CsPV. (**A**) Phylogenetic tree of papillomaviruses based on the aligned amino acid sequences of the CsPaV L1 protein and the L1 proteins of other fish papillomaviruses. (**B**) Phylogenetic tree of the parvovirus based on the amino acid sequence alignment of the CsPV NS1 protein and the NS1 proteins of other representatives of the ten genera in the subfamily Parvovirinae, and two unassigned fish parvoviruses in the subfamily HamaParvovirinae. The phylogenetic trees were constructed using the maximum likelihood method, with 100 bootstrap replicates. The scale bar means the genetic distance, number of substitutions per site.

**Figure 5 viruses-16-00705-f005:**
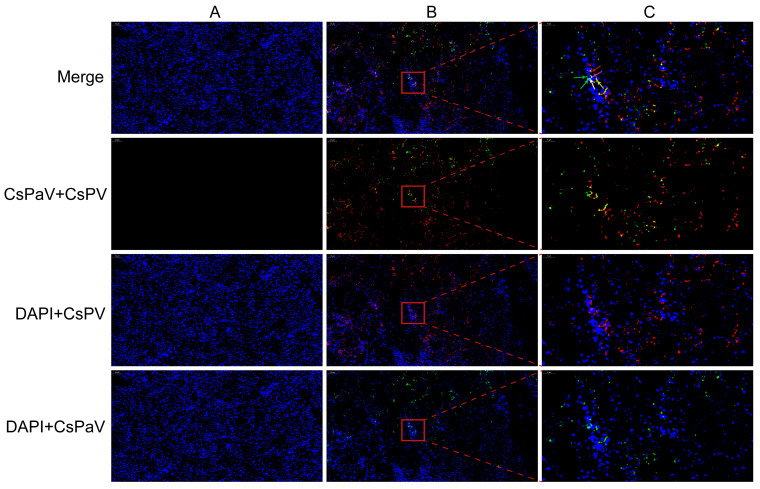
Detection of the mRNAs of CsPaV and CsPV in the infected spleen of Chinese tongue soles using FISH. (**A**) Uninfected spleen (Bar, 50 μm); (**B**) Infected spleen (Bar, 50 μm); (**C**) High magnification of the red rectangular region (Bar, 10 μm). Tissue sections display red fluorescence for CsPV (Cy3-conjugated probes), green for CsPaV (FAM-conjugated probes), and blue for cell nucleus (DAPI). The mRNAs of CsPV and CsPaV were detected in cytoplasm, which showed red (red arrow) and green (green arrow), and the nucleus showed purple (purple arrow) and cyan (cyan arrow). The mRNAs of the two viruses co-located in the cytoplasm showed yellow (yellow arrow), and the nucleus showed white (white arrow) in one cell.

**Figure 6 viruses-16-00705-f006:**
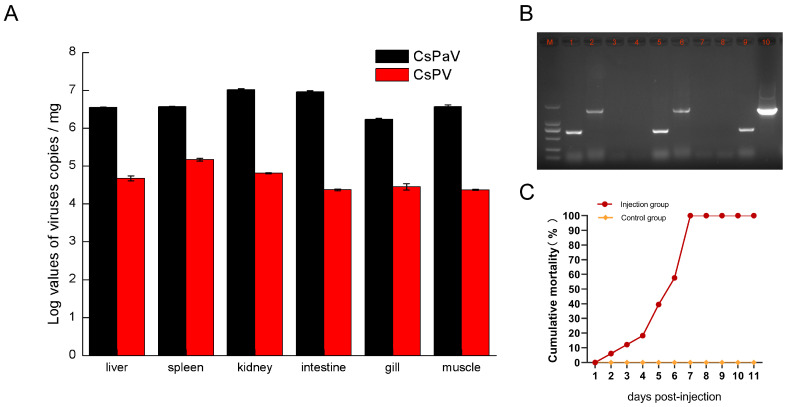
Detection and distribution of CsPaV and CsPV in diseased Chinese tongue sole. (**A**) Distribution and copy numbers of CsPaV and CsPV in the liver, spleen, kidney, intestine, gills, and muscle. Data are the means and standard errors of the means for three replicates. (**B**) PCR detection of CsPaV and CsPV in the experimental group and control group. Lane M: DL2000 Marker; Lane 1 and 2: CsPaV and CsPV in homogenate for injection; Lane 3 and 4: CsPaV and CsPV in the kidney sample of control group; Lane 5 and 6: CsPaV and CsPV in kidney sample of experimental group; Lane 7 and 8: negative control of CsPaV and CsPV; Lane 9 and 10: positive control of CsPaV and CsPV. (**C**) Cumulative mortality of Chinese tongue soles infected with CsPaV and CsPV.

**Table 1 viruses-16-00705-t001:** Primers and probes used in the present study.

Primer or Probe	Sequence (5′-3′)	PCR Product (bp)	Purpose
PaV-F2	ATTAGAGCACGCCATTCA	1147	Overlap PCR to amplify the genome of CsPaV
PaV-R2	TCAGTAAGATATATAATAGC		
PaV-F3	GCTATTATATATCTTACTGA	661	
PaV-R3	CCCTTCTTCTAAATCCCT		
PaV-F4	AGGGATTTAGAAGAAGGG	2159	
PaV-R4	GTCATCCTTACTATATCGCC		
PV-F1	AAATCTTCGCCAGGTA	231	Overlap PCR to amplify the genome of CsPV
PV-R1	CCAAGAGATGATACCC		
PV-F2	CAAAAAACCCTCCTCCA	452	
PV-R2	GTTTCGTTTGCC		
PV-F3	AGAACAGGCAAACGAAAC	601	
PV-R3	CCTGTGTGCAGACGAGCT		
FISH-L1	GGACTGCCACTGGTATCTTCCTTGA, GTTCACTGAGATTTCCCCCTTCTG, TGCGGTTTCAGTAACATATTCATCG, CCTTGGAGAGGTGGCAAGTAGAATT, GGATCTGTTTATGTTCAATTCCCCA		Probes used in FISH for detection of CsPaV
FISH-VP	GATTGGCCGCTAGTATCAGGTTTTC, AGGGTTACCCATTTCGGTTTTGTAA, GGGCGCCATTTTTCATAGTGATTTA, CAATGGCAAGAGCAGCAGTTAAATC, GTGTATCTTGCAGCGCTTGTCTTTT		Probes used in FISH for detection of CsPV

## Data Availability

The sequencing reads were deposited into the Sequence Read Archive (SRA) under accession number PRJNA1040012.

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
