# Peer review of "Determination and Characterization of Novel Papillomavirus and Parvovirus Associated with Mass Mortality of Chinese Tongue Sole (Cynoglossus semilaevis) in China"

_viruses, 2024, doi:10.3390/v16050705_

Round 1

Reviewer 1 Report

Comments and Suggestions for Authors

Xue et al. discovered and characterized two highly virulent new genera of papillomaviruses and parvoviruses isolated from diseased Chinese tongue soles (CsPaV and CsPV, respectively), that caused high mortality during an outbreak in December 2021 in a commercial fish farm in Tianjin, China.
The authors applied a very thorough and remarkably well-designed approach to amplify, sequence, and genetically characterize the two novel viruses, identifying CsPV as the first known disease-causing parvovirus identified from marine cultured flatfish.
These viruses represent potential novel viral pathogens of farmed Chinese tongue sole, urging the need for future studies to address pathogenesis mechanisms of these two viruses, as well as investigate the feasibility of possible control measures to limit the spread of these pathogens to other farms of this fish species (as well as potentially to other species in the future).
This study has been thoughtfully planned and executed, and the writing is clear and effectively communicates the research findings. However, there are a few minor points to correct and questions to address before considering this manuscript for publication in Viruses.

- line 92 – replace “absorption” with "adsorption".
- line 101 - Replace "Gequencing" with "Sequencing".
- Line 133 - "lipo3000" should be replaced with the complete product name (Lipofectamine 3000), and acknowledge the company making it.
- Line 170-171: 35 cycles of 95°C for 5 min, 94°C for 1 min, 58°C for 1 min, and 72°C for 1 min (...) - the initial denaturation (95°C for 5 min) is performed only once, not 35 times. The same in line 176 (35 cycles of 95°C for 3 min). Please correct.
- Lines 191-192: The virus copies were calculated according to the Ct values and the formula generated above (...) - Where is the formula? (I didn’t see it).
- Line 206 - cumulative mortality reached 80.7% (Figure 1B). Clinical symptoms observed included punctuate hemorrhages in the liver, a swollen spleen, and kidneys filled with white nodules (Figures 1A-1 and A-2) - did the authors observe similar lesions in fish that survived? Were fish that survived analyzed at all?
- In figure 1, you should add images of organs and histological images of mock-infected fish (or fish showing no disease symptoms) for comparison, to better appreciate the differences between a normal organ structure and pathological changes caused by the two viruses.
- Fig. 2: what was the highest viral titer obtained at the second cell culture passage at 8 dpi in FG cells?
- Line 348 - instead of "virus loading", I believe the authors intended "virus load"?
- Line 349-350 - please specify that the values refer to viral genomic copies. Also, please add for example the exponent symbol (^) to all your numbers to clarify that you are talking about Log values (10^7.02 instead of 107.02?).
- Line 354 - replace "The virus copies of CsPV" with "The virus genomic copies of CsPV".

Reviewer 2 Report

Comments and Suggestions for Authors

Article title: Determination and characterization of novel papillomavirus and parvovirus associated with mass mortality of Chinese tongue sole (Cynoglossus semilaevis) in China. 

In this paper, the authors have identified two Novel viruses. One named as Cynoglossus semilaevis papilloma virus (CsPaV) and other Cynoglossus semilaevis parvovirus (CsPV) from Chinese tongue soles (Cynoglossus semilaevis). For this, the authors used transmission electron microscopy to observe virus particles, 

They isolated and propagated the virus in the flounder gill cells. The authors sequenced their genomes and performed an experimental in vivo challenge using the same host. At this point, I wondered whether they checked that fish were negative prior to the in vivo challenge. In addition, were they the same age as the diseased fish in the first place?. This should be specified in the M&M section. 

Sequences of L1 and NS1 revealed that they were novel members of the previously mentioned virus families…Are these segments usually used in the literature to phylogenetically classify papilloma y parvovirus? Why is L1 specified only in M&M s? 

They reported that following FISH results, they detected the virus in the spleen of infected fish and that viruses could co-infect single cells. Is this common occurrence for these viruses ? Do they usually allow cells to be co-infected or superinfected with viruses from different families? If so, they should explain this further in the Discussion section. 

The authors start mentioning that massive mortality occurs, but they do not specify the clinical or pathological signs that are characteristic of a possible disease. What happened during the in vivo trial ? Did they reproduce the same disease that occurs in farmed Chinese tongues?

Materials and Methods: 

The authors did not specify what clinical or macroscopic pathological signs were characteristic of diseased fish and what clinical signs were detected in the in vivo trial separately. Were they the same? Did they vary in intensity?. However, this should also be specified.

Did any of the control animals display any macroscopic pathological signs? 

The age of the animals should be specified; the authors only provided the length. In vivo trials of diseased animals of the same age 

Electron microscopy did not reveal any details of the negative staining performed on the samples. This makes reproduction more difficult. 

Is the citopatic effect similar to that of other papillomas or parvoviruses? Is this specified in the introduction ? 

Were are the primers taken from ? Did they contrast these sequences with those of other papillomas or parvoriviruses from databases?

In the in vivo trial, they did not use any contact (non-infected) animals in the same group. This appeared to be an in vivo passage. However, I wondered how the authors could be sure that the virus was actually transmitted from infected to uninfected animals. Could they detect the same inoculum they are using in the first place? 

In the discussion, it is not clear why the authors initially looked for these viruses in the first place. Did you look for any bacteria first ? What led them to try these two viruses? 

Why did they use FG cells ? Do the authors believe that the virus can infect other cell cultures, such as CHSE-214, RTG-2, or other commonly used cell cultures? Is there any literature in this respect, or have they tried to use a different cell culture? 

Reviewer 3 Report

Comments and Suggestions for Authors

In this manuscript Xue et al identified two new viruses in farmed Chinese tongue soles during an outbreak that occurred in Tianjin. These two novel viruses, Cynoglossus semilaevis papillomavirus (CsPaV) and parvovirus (CsPV) were simultaneously isolated and identified from diseased fish by electron microscopy, virus isolation, genome sequencing, experimental challenge and fluorescence in situ hybridization (FISH). Electron microscopy showed large numbers of virus particles present in tissues of diseased fish whereas viruses isolated and propagated in flounder gill cells (FG) induced typical cytopathic effects (CPE). The FISH results showed signals in spleen tissues of infected fish, and both viruses could co-infect single cells The complete genomes of CsPaV (5939 bp and CsPV 3663 bp) shared no nucleotide sequence similarity with other viruses. Phylogenetic analysis based on the L1 and NS1 protein sequences showed that CsPaV and CsPV were

novel members in the Papillomaviridae and Parvoviridae families.

The manuscript is well written, and the results presented represent an interesting contribution to the field of fish viral pathogens. Techniques utilized are extensively described and the isolation, tissue distribution and experimental infection studies confirm the role of these agents in this disease. The authors also discuss the limitations of the study in the identification of which of the two viruses is responsible for the disease, question that require further studies although it is not atypical the simultaneous infection of parvoviruses with others viral agents.

For this reviewer, there are no major suggestions to make except to explain the choice of the E1 gene for the realtime assay for papillomavirus (the point 2.9).

Comments on the Quality of English Language

The English writing is quite good, a minor editing could be beneficial
